# SDHAR-HOME: A Sensor Dataset for Human Activity Recognition at Home

**DOI:** 10.3390/s22218109

**Published:** 2022-10-23

**Authors:** Raúl Gómez Ramos, Jaime Duque Domingo, Eduardo Zalama, Jaime Gómez-García-Bermejo, Joaquín López

**Affiliations:** 1CARTIF, Technological Center, 47151 Valladolid, Spain; 2ITAP-DISA, University of Valladolid, 47002 Valladolid, Spain; 3System Engineering and Automation Department, EEI, University of Vigo, 36310 Vigo, Spain

**Keywords:** dataset, deep learning, sensors, activity wristbands, beacons, neural network, smart home

## Abstract

Nowadays, one of the most important objectives in health research is the improvement of the living conditions and well-being of the elderly, especially those who live alone. These people may experience undesired or dangerous situations in their daily life at home due to physical, sensorial or cognitive limitations, such as forgetting their medication or wrong eating habits. This work focuses on the development of a database in a home, through non-intrusive technology, where several users are residing by combining: a set of non-intrusive sensors which captures events that occur in the house, a positioning system through triangulation using beacons and a system for monitoring the user’s state through activity wristbands. Two months of uninterrupted measurements were obtained on the daily habits of 2 people who live with a pet and receive sporadic visits, in which 18 different types of activities were labelled. In order to validate the data, a system for the real-time recognition of the activities carried out by these residents was developed using different current Deep Learning (DL) techniques based on neural networks, such as Recurrent Neural Networks (RNN), Long Short-Term Memory networks (LSTM) or Gated Recurrent Unit networks (GRU). A personalised prediction model was developed for each user, resulting in hit rates ranging from 88.29% to 90.91%. Finally, a data sharing algorithm has been developed to improve the generalisability of the model and to avoid overtraining the neural network.

## 1. Introduction

In recent years, recognising the activities performed by people has become a very important goal in the field of computing and DL [1]. Achieving this goal means an improvement in various fields, such as security, health, welfare and surveillance. One of the main difficulties of this research is to recognise activities with a high success rate in real environments. In addition, the objective is also to achieve this goal with a low computational load [2]. This type of system can be used to improve the quality of life and increase the care and safety of elderly people [3] and people with a certain level of dependence [4].

Recognising the activities carried out during the day is an expensive and difficult task, both from a technological point of view and a social perspective due to the privacy question and the exploitation of these data [5]. It is necessary to take into account the number of different domestic activities that a person performs from the time that he/she gets up in the morning until he/she goes to bed at night. For example, a study carried out by the American Time Use Survey (ATUS) in 2020 reports that the inhabitants of the United States of America (USA) spend an average of 1.7 h doing domestic activities compared to only 18 min a day doing sports or leisure activities [6]. Therefore, the scientific community is trying to publish different datasets with labelled activities in order to provide information on different situations in which different domestic activities are being performed. However, the current datasets present limitations in the number of categories, samples per category or temporal length of the samples [7]. In addition, another major challenge is the fact that multiple residents can cohabitate within the same household, which complicates the recognition task. For example, a family of several members can live together in the same house and perform different tasks at the same time [8]. This implies a need to improve the technology to be used for detection or the need to record a larger variety of events inside the house in order to cover as many cases as possible [9].

Several solutions have been proposed towards recognizing people’s daily life activities in their homes. Some are based on video recognition, using RGB cameras placed in different locations of the house [10]. Other solutions propose the use of audio collected through a network of microphones, since it is possible to classify activities based on the analysis of audible signals [11]. However, this type of solution is generally rejected by the users, since it compromises their privacy at home to a great extent [12]. Other solutions aim to recognise activities by analysing the signals provided by wearable sensors, such as activity bracelets or the signals provided by the smartphone [13]. However, databases developed using this type of solution only provide information about activities from a physical point of view, such as walking, running, standing up or climbing stairs. Actually, the most interesting activities that people can perform at home, and the most interesting to detect, are those performed daily that provide valuable information about everyday habits [14]. These daily habits can provide information of vital importance, such as whether the person is correctly following the medication guidelines or whether they are eating correctly and at the right times of the day [15].

The main purpose of this paper is to create a database that collects information about the daily habits of multiple users living in the same household through the implementation and cooperation of three different technologies. In addition, this paper establishes a baseline research on the recognition of the activities of these users, through the implementation of different Deep Learning methods using the information from the proposed database. Different DL methods that can create temporal relationships between the measurements provided by the monitoring system are proposed, as it is not possible to create a static model using simple automated rules due to the complexity and dependencies of the model.

The main contributions of the paper are the following: (i) A public database has been created that has been collecting information on the main activities that people carry out in their homes (18 activities) for two months and which is available for the use of the scientific community. (ii) The database has been constructed in an environment where two users, who receive sporadic visits, live with a pet. (iii) The technology used for the elaboration of the database is based on the combination of three different subsystems: a network of wireless ambient sensors (e.g., presence detectors or temperature and humidity sensors) that capture the events that take place in the home, a positioning system inside the house based on Bluetooth beacons that measure the strength of the Bluetooth signal emitted by users’ wristbands, and the capture of physical activity and physiological variables through the use of wearable sensors. To the best of our knowledge, no database currently exists in the scientific community that simultaneously incorporates all three types of technology. (iv) The technology used to detect the activities is non-intrusive, thus guaranteeing the well-being of the users inside the house while preserving their security and privacy. (v) In order to set a baseline, different DL techniques based on recurrent neural networks have been used in order to carry out real-time activity recognition for both users to know their daily habits. These DL techniques provide hit rates ranging from 88.29% to 90.91%, depending on the user.

The article is structured as follows: Section 2 analyses several recognized databases to perform the recognition of human activities in daily life, as well as some representative activity recognition methods. Section 3 details all the information relevant to the database that includes the technology used, in addition to the method of data collection and storage. Section 4 analyses the different Deep Learning techniques used to perform the activity recognition, as well as the method of analysis and data processing and the network architecture proposed as the research baseline. Section 5 details the different experiments carried out with the data extracted from the proposed database. Finally, Section 6 summarises the strengths and conclusions of the research and suggests future research lines.

## 2. Related Work

In this section, the main public datasets for activity recognition and the latest technologies to perform this task are described.

### 2.1. Summary of Database Developments

There are multiple databases in the scientific community designed to perform activity recognition, with different technologies, dimensions and number of activities. For example, the USC-HAD [16] dataset was developed in 2012, and the technology used is based on wearable sensors. This database collects information on physical activities, such as sitting on a chair, walking or running. In total, there are 12 different labelled activities for multiple users, of different gender, age, height and weight. However, although physical activities are important, there are other activities that are performed during the day that have a great influence on people’s habits and quality of life. Another interesting dataset is the MIT PlaceLab Dataset [17], which combines different technologies to take measurements, such as ambient light sensors, proximity sensors or pulse sensors. This dataset was created in 2006 and was developed under non-ideal conditions, i.e., outside of a controlled laboratory. The dataset has a single study subject and a total of 10 labelled activities. The labelled activities correspond to 10 activities of daily living, such as making the bed or cleaning the kitchen. However, information on many other important daily activities is lost. The ContextAct@A4H database [18] is composed of non-intrusive sensor data obtained in real life situations of a 28 year old woman living in a flat for 4 weeks. It has a total of seven labelled activities and a total of 219 sensors and actuators. The SIMADL database [19] collects information on seven people living non-simultaneously in a household with 29 binary sensors for 63 days, recording both their activities and various anomalies that may occur in the household, such as leaving the TV on or the fridge open. Participants tag five generic activities (eating, sleeping, working, leisure or personal). Another interesting database could be MARBLE [20]. In this database, the authors collect information from 8 environmental sensors and the information published by a smart watch. The data collection is carried out with 12 different users in individual cases, performing 13 different activities during 16 h. The OPPORTUNITY [21] dataset is widely used within the scientific community. In this dataset, 72 sensors of 10 different modalities have been deployed in 15 different installations, in order to obtain data from 12 different users who do not live together. Typically, studies of this dataset have a duration of 25 h of data. However, a 25 h recording process can hardly provide information about a person’s daily habits. Another interesting project is the one carried out by the University of Amsterdam (UvA) in 2008 [22], with a 28-day deployment in which a total of 2,120 events were collected from 14 sensors located on doors, the refrigerator, etc., to study the activity of a 26-year-old adult. However, this project provides information on a limited number of events. The CSL-SHARE [23] database hosts data provided by wearable sensors (10 sensors) to perform the recognition of 22 physical category activities, such as sitting, standing up or running. In the data collection phase, 20 different users who do not live together are monitored by the authors for 2 h while they perform various executions of these activities. This is useful to detect physical activities, but with such a short duration it cannot provide information on daily habits. Other authors decided to create datasets of images and videos in order to carry out recognition using convolutional neural networks (CNN). This is the case of the NTU RGB+D [24] dataset, in which several images of users performing activities are collected. Recognition is addressed by combining the images with the position of the user’s joints. In total, 60 different types of activity are analysed, in which 40 classes correspond to daily life activities, 9 to health-related activities (such as falling or sneezing) and 11 to interactions between users, such as hugging or kissing. To take the videos and photos, the authors used a Kinect v2 camera. However, the use of this type of technology is not recommended as it compromises privacy. With respect to datasets dealing with multiple cohabiting users, fewer works can be found in the literature. One example is the “Activity Recognition with Ambient Sensing” (ARAS) dataset [25], focused on collecting information from two different real houses with multiple cohabiting residents for a total duration of two months. In total, 27 different activities have been tagged using information collected by 20 binary sensors communicating over the ZigBee protocol. Another dataset based on the collection of data from multiple residents is CASAS [26] project, developed in 2010, in which different databases have been made depending on the house and the situations, with one or more residents living together. These residents have diverse ages and, therefore, different habits. 11 activities are labelled for a variable duration, ranging from 2 months to 8 months, depending on the database analysed [27]. With respect to both projects, without including a technological mechanism capable of discerning which person is carrying out the activity or their location, it is not possible to know which of the users is carrying out the activity. The FallAIID dataset [28] includes 26.420 data files corresponding to the measurements of different gyroscopes that users carry while falling. The Activities of Daily Living (ADL) dataset [29] is composed of measurements from gyroscopes worn by 10 volunteers while performing 9 types of activities of daily living, and at least 14 samples of each activity. Another database that uses inertial sensors is Cogent-House [30], in which both domestic activities and falls are labelled. In this database 42 volunteers have participated, 4 types of falls and 11 activities of daily living have been simulated. The dataset is composed of data provided by 12 inertial sensors. There are databases that collect the information provided by the sensors of users’ smartphones. For example, the database developed by Pires, I. et al. [31] collects information on 25 users performing 5 physical activities during 14 h. However, real-life users do not carry their smartphones 24 h a day, so this data collection is not a true representation of reality.

In short, the existing databases often do not have a sufficient number of samples to train a DL model. Moreover, there is a limited variety of databases with labelled activities for several cohabitants. Besides, some databases have quite a reduced range of activities of daily living, such as eating, sleeping, cooking or personal hygiene. Furthermore, the databases that include such activities often do not use any technology able to discern which user is performing the activity. The database proposed in this paper has a significant number of labelled activities of daily living for multiple users, over a significant time range and using a combination of three different technologies: a non-intrusive wireless sensor network, a positioning system using Bluetooth beacons and a wearable sensor-based user monitoring system. The proposed database allows the users’ healthy lifestyle habits to be monitored, and serves to prevent risky situations, in particular for older people.

ARAS [25] is probably the most similar database to the one proposed in this paper, as both contain data from a house in which two users cohabit for a period of two months. The ARAS authors perform activity recognition by modelling activities and data sequences using a Hidden Markov Model (HMM) and cross-validation. The authors obtain an average accuracy of 61.5% for House A and 76.2% for House B concerning the activity recognition. These accuracy rates are lower than those obtained in the present paper. In contrast, the database proposed in the present paper includes a larger number of ambient sensors, a system that captures the position of each user within the house and physiological information of each user, which can help with the recognition of activities. In addition, positioning technology makes it possible to discern the user’s position in the home in order to differentiate which person is performing the activity. CASAS [26] is also very similar to the database developed in the present paper as it has multi-user information and uses environmental sensing technology and wearable devices. However, the position of the users at home is not discriminated and some activities are not considered (e.g., watching tv, reading or taking meds). An overall hit rate of activity recognition of 84.14% is achieved through Support-Vector Machines (SVM) algorithms, while a higher score is obtained in the present paper. Furthermore, although their overall hit rate is adequate, the hit rate for a few specific activities (e.g., eating or relaxing) is lower than 50%.

A global overview of the databases analysed in this section can be found in Table 1.

### 2.2. Summary of Activity Recognition Methods

Many authors are currently working on the development of algorithms to exploit the information from the databases mentioned in Section 2.1 towards achieving systems able to recognise activities to a high hit rate. A representative example is the work carried out by Ramos, R.G. et al. [32], in which data from a CASAS database is used to develop a real-time recognition system with an accuracy rate of 95.42%. To achieve this accuracy, the authors use DL methods based on recurrent LSTM neural networks [33], fed through a sliding window of contiguous data equispaced in time. A good comparison with this work is that developed by Liciotti, D. et al. [34], whose algorithm achieves an accuracy of 93.42% for the same database. Another example is the work by Xia, K. et al. [35], in which a number of different systems based on LSTM networks and CNN networks are developed and tested on the OPPORTUNITY dataset, leading to over 90% accuracy. It is possible to recognise the activities carried out by residents using cameras to track their body joints during the execution of activities [36]. For example, the work developed by Imran, K.U. et al. [37], by implementing the combination of CNN and LSTM networks, the authors are able to recognise 12 physical activities thanks to the spatial positions provided by a Kinect V2 sensor camera. Another example could be the skeleton tracking integrated with the Multilayer Perceptron (MLP) and HMM used in the work developed by Domingo, J.D. et al. [38], in which the authors manage to recognise exercises performed during gymnastics sessions. Their system is able to recognise a total of 19 postures through computer vision by means of frame analysis, reaching a recognition success rate of 98.05% in the best of situations. Another method of recognition using different technology is based on the use of microphones that register noise related to the activity being performed [39]. The work by Yong Li and Luping Wang [40] focuses on the performance of a neural network based on bidirectional LSTM layers combined with residual blocks. The authors evaluate their method using data from the WISDM and PAMAP2 databases, which store accelerometer data, and they build their own database by collecting information from 6 physical activities. The authors achieve a success rate of over 90%, but it is necessary to consider that the types of activity they are able to detect are of a physical nature, and do not obtain information about the daily habits of a person at home. Other authors use the information collected by a smartphone to detect activities [41]. For example, the work developed by Wan, S. et al. [42] shows the effectiveness of 5 different DL models (CNN, LSTM, bidirectional LSTM, MLP and SVM) applied to PAMAP2 and UCI datasets, giving results of approximately 90% accuracy. However, these methods are not comparable to the dataset proposed in this paper, as they do not detect activities corresponding to the daily habits of a person. The work developed by Zolfaghari, S. et al. [43] focuses on the development of a vision-based method to obtain user trajectories inside homes by using data obtained from environmental sensors, specifically from CASAS database. With this feature extraction method, the authors obtain accuracy rates close to 80% by using different machine learning algorithms, such as SVM or MLP. The TraMiner prototype [44] is also capable of recognizing locomotion patterns inside homes. In order to develop this prototype, authors have used information from the trajectories of 153 elderly people, including people with cognitive problems. This information is also contained in CASAS database.

## 3. The SDHAR-HOME Database

This Section explains the development of a sensor dataset for human activity recognition at home (SDHAR-HOME) and the different subsystems that are in charge of generating the data. A general diagram can be seen in Figure 1.

First, the use of non-intrusive sensors to record the different events that take place inside the home will be explained. Then, the use of Bluetooth beacons to obtain the position of each user is described. The use of activity wristbands to obtain data linked to each user, such as pulse, steps or hand position, will also be discussed. Finally, the use of an InfluxDB database to collect all the information generated by the aforementioned subsystems is explained, as well as the concept and functioning of such database.

The database collection has been carried out in a house where two people are living together with a pet. In addition, they receive sporadic visits. In order to build the database, measurements were taken over a period of 2 months. During this period, possible changes in moving objects or changes in network conditions have been considered. All these events provide value to the database, reproducing in an integrated way different conditions that would happen in real life. The circumstance of receiving visitors is a common occurrence in everyday life, the same as having a pet in the home. The alteration of sensor measurements due to these visitors and the pet is assumed as noise that the recognition method has to filter out, given that the recognition of their activities is not sought. Table 2 shows the activities labelled in the database by both users. It should be noted that each activity is independent, and only one activity can be carried out by each user at a given time. The activity “Cook” is tagged when one of the users is making an elaborate meal, in which it is necessary to use the ceramic hob and usually produces sudden changes in temperature near the ceramic hob. On the other hand, the “Make Simple Food” activity is a basic meal that takes little time, such as preparing sandwich or making a salad. The “Pet” activity corresponds to basic pet care, and the “Other” activity corresponds to the time when none of the other activities in the database are being carried out.

The activities that make up the dataset have been tagged by the users themselves during their daily lives, using a set of NFC tags. Each user passes their smartphone over an NFC tag when carrying out an activity. Then, an application deployed in the smartphone sends a message to the operating system used in the proposed solution, which records the activity and the current time. Daily, one of the users is responsible for checking all manually tagged activities, in order to make corrections for both users if it is necessary.

In order to determine the number of sensors needed to detect all the activities and their type, a preliminary study was carried out on the main characteristics of each activity and how to detect them. Table 3 shows the rooms of the house, as well as the distribution of activities that are most frequently carried out in these rooms. In addition, the table shows the minimum number of sensors to detect each activity as well as their location in the room. It is estimated that, in order to establish a research baseline for activity recognition using DL techniques, information from the non-intrusive sensors and the position provided by the Bluetooth beacons is needed. However, the database includes information from more sensors that could be used in further developments to improve the performance of the activity recognition system.

### 3.1. Overview of the Monitoring System: Hardware, Software and Control

The three sub-systems emit information simultaneously, which needs to be collected by a central module that acts as the system controller. For this purpose, an NVIDIA Jetson Nano computer was chosen due to its suitable processing power. In addition, this computer has a GPU where both the training sessions and the execution of the neural network that processes the data and obtains the users’ activities in real time can be done. The Home Assistant operating system [45] has been installed on the controller to manage the data collection and storage.

This operating system is responsible for linking each sensor to the controller in order to record the flow of information. In addition, a visual interface is provided where it is possible to verify that the system is working correctly. The visual interface developed for the non-intrusive monitoring system can be seen in Figure 2.

In this visual interface, both the non-intrusive sensor subsystem and the position of each of the users within the house can be visualised. The position is obtained by triangulation using Bluetooth beacons. Each user wears a smartband, and the monitoring system developed in this paper is responsible for recording the strength of the Bluetooth signal to each one of the beacons deployed in the home. Therefore, in order to determine which room the user is in, the system chooses the beacon that receives the strongest signal. In the interface, if the users move through each of the rooms of the house, the symbol assigned to each user also moves around the map. This is conducted by placing the user’s symbol in the room whose beacon provides the nearest distance. In addition, the visual interface allows the time evolution graphs of each sensor to be viewed. Finally, this interface also provides real-time visualisation of the information collected by the activity wristbands, such as the pulse or the steps taken by each of the users.

The data collected by the Home Assistant are exported in real time to a database located on an external server in order to avoid the saturation of the main controller. It is worth mentioning that the data generated by the gyroscopes of the activity wristbands are not brought to the Home Assistant, as the service would quickly become saturated due to the high flow of information that they generate (multiple samples per second). Instead, this information is taken directly to the external server database.

### 3.2. Non-Intrusive Sensor-Based Home Event Logging

A type of non-intrusive technology was chosen to generate the dataset, which collects information on the events that take place inside the house: door opening, presence in a certain room, consumption of a household appliance or variation of the temperature and humidity at certain points in the house. This type of technology does not compromise the privacy of users as no information intimately linked to the person is recorded. Table 4 shows the set of sensors used in the installation, their total number and their reference.

The use given to each sensor, as well as the location where it is placed, is detailed below:Aqara Motion Sensor (M): This type of sensor is used to obtain information about the presence of users in each of the rooms of the home. Six sensors of this type were initially used, one for each room of the house. In addition, two more sensors were used to divide the living room and the hall into two different zones. These two sensors were located on the walls, specifically in high areas facing the floor.Aqara Door and Window Sensor (C): This type of sensor is used to detect the opening of doors or cabinets. For example, it is used to detect the opening of the refrigerator or the microwave, as well as to recognise the opening of the street door or the medicine drawer. In total, eight sensors of this type were used.Temperature and Humidity Sensor (TH): It is important to know the variations of temperature and humidity in order to recognise activities. This type of sensor was placed in the kitchen and bathroom to recognise whether a person is cooking or taking a shower, as both activities increase the temperature and the humidity. These sensors were placed on the ceramic hob in the kitchen and near the shower in the bathroom.Aqara Vibration Sensor (V): The vibration sensor chosen is easy to install and can be hidden very easily. This sensor perceives any type of vibration that takes place in the furniture where it is located. For example, it was used to perceive vibrations in chairs, to know if a user sat on the sofa or laid down on the bed. It is also able to recognise cabinet openings.Xiaomi Mi ZigBee Smart Plug (P): There are activities that are intimately linked to the use of certain household appliances. For this reason, an electrical monitoring device was installed to know when the television is on and another consumption sensor to know if the washing machine is being used. In addition, this type of sensor provides protection against overcurrent and overheating, which increases the level of safety of the appliance.Xiaomi MiJia Light Intensity Sensor (L): This sensor detects light in a certain room. This is useful when it is necessary to detect, for example, if a user is sleeping. Presence sensors can recognise movement in the bedroom during the day, but if the light is off, it is very likely that the person is sleeping. This sensor was also used in the bathroom.

The sensors mentioned above, from Aqara [46] and Xiaomi [47], are able to communicate using the Zigbee protocol [48], which is a very low-power protocol because its communication standard transmits small data packets. This enables the creation of a completely wireless network, which makes installation and maintenance very versatile (the sensor battery only needs to be changed when it runs out of power and lasts for more than a year of continuous use). The sensors emit Zigbee signals that need to be collected through a sniffer supporting this type of communication that transforms it into a signal known by the controller. To this end, a ConBee II sniffer was chosen that is responsible for transforming the Zigbee signals into Message Queue Telemetry Transport (MQTT) messages that are supported by the controller and can be managed by the Home Assistant. The MQTT protocol is a machine-to-machine protocol with message queue type communication created in 1999, which is based on the TCP/IP stack for performing communications. It is a push messaging service with a publisher/subscriber (pub-sub) pattern which is normally used to communicate low-power devices due to its simplicity and lightness. It has very low power consumption and requires minimal bandwidth and provides robustness and reliability [49].

### 3.3. Position Triangulation System Using Beacons

The use of non-intrusive sensors that capture events occurring in the home does not guarantee good activity recognition for each user. Using only the information from the sensors, it is impossible to know which user performs the action. Therefore, in order to classify the activity and identify the position of each user inside the house, a number of Bluetooth beacons were installed in each room of the house [50]. These beacons measure the Bluetooth power emitted by a wearable device such as an activity wristband. In this way, it can be ensured that the user will be wearing the device most of the time (except when it is necessary to charge its battery). Therefore, Xiaomi Mi Smart Band 4 devices were chosen as it is a widely marketed and affordable device.

The Bluetooth beacons were developed on an ESP32 chip, as it is a cheap microcontroller and increasingly used. This microcontroller has a number of advantages over other solutions, such as the fact that it incorporates WiFi, Bluetooth and Bluetooth Low Energy (BLE) connectivity [51]. The software used in the ESP32 is developed by the ESPresense group [52]. This development is easy to deploy, uses a fingerprint instead of a fixed MAC address, which facilitates the incorporation of Apple-IOS devices and applies a Kalman filter in order to reduce jitter. The beacons measure the Bluetooth power of the connection with the activity wristbands and publish its value through MQTT messages that are collected by the main controller. The topics of the MQTT messages contain the measuring device and the beacon publishing the message, so that the controller can use this information.

### 3.4. User Data Logging via Activity Wristbands

The information collected by the activity wristbands selected for this research provide useful parameters linked to the physical activity of the users. These wristbands incorporate gyroscopes and accelerometers on all three axes of the Cartesian system, and are capable of measuring heart rate through photoelectric sensors [53]. Therefore, they are able to provide the following information:Heart rate: The wristbands estimate the heart rate through the difference between a small beam of light emitted at the bottom and the light measured by a photoelectric sensor in the same position. In this way, the heart rate per minute can be extrapolated from the amount of light absorbed by the wearer’s blood system.Calories and fat burned: Thanks to measurements of the user’s heart rate, and supported by physiological parameters provided by the user, the wristbands are able to estimate calories and fat burned throughout the day.Accelerometer and gyroscope data: Wristbands do not broadcast information about internal accelerometers and gyroscopes, as this would result in a premature battery drain due the huge information flow. Instead, the emission protocol of the wristbands has been modified to share this information, as the position and movements of the hand are very important in the execution of activities.Steps and metres travelled: In the same way as calories and fat burned, using the information from accelerometers and gyroscopes, with the support of physiological parameters provided by the user, it is possible to estimate the steps and metres travelled by the user at any time.Battery status: The wristband also provides information on its battery level, as well as its status (whether it is charging or in normal use).

The standard connection of the wristbands is based on a Bluetooth connection between the band and a smartphone. Using this protocol, the user’s data are communicated to the Xiaomi cloud. For this reason, it is necessary to incorporate an intermediary between the wristband and the smartphone to collect the data and incorporate them into the database of the system. Therefore, since the main controller of this system (NVIDIA Jetson Nano) does not include Bluetooth connectivity, a Raspberry Pi 4 was used as a secondary controller that does support this type of communication [54]. Moreover, it emits a BLE message, which orders the wristband to start emitting the information described above, and is also responsible for storing the gyroscope data locally and publishing the rest of the information to the main controller in real time via MQTT messages. Finally, once a day, the gyro information is sent to the external database (usually at night, as this is the least busy time in the house) and the information is deleted locally to free up space in the board’s capacity.

### 3.5. Data Storage in InfluxDB

All data collected by both the main controller (NVIDIA Jetson Nano) and the secondary controller (Raspberry Pi 4) are sent to a database mounted on an external server to ensure data security. An InfluxDB database was chosen to collect and organise the data due to the following advantages [55]: it is an open source database that is optimised for processing time series and large data streams, it provides support for a large number of programming languages (such as Python, R, etc.), it offers support for Grafana that enables the creation of customisable visualisation dashboards, it is easy to install and there is extensive documentation on its use and maintenance [56]. The Grafana tool were used to visualise the history of the data that conforms the database. Grafana is an open source tool used for the analysis and visualisation of user-customisable metrics. The user can customise panels and create custom dashboards, giving the user the ability to organise the data in a flexible way [57].

Figure 3 shows the communication paths between the different devices used in the developed solution. It can be seen how all the data reach the InfluxDB database and can be visualised using the Grafana tool.

## 4. Activity Recognition through Deep Learning

In this section, the system proposed to detect the activities carried out by users inside their homes, based on the data obtained from the system proposed in Section 3, is described. First, conventional DL methods are described, including different RNNs such as LSTM networks for dealing with time series data, or GRU methods that represent an advance with respect to LSTM. Then, the different data processing methods carried out in the present system are explained. Finally, the resulting architecture will be shown.

### 4.1. RNN-Models

Models based on recurrent neural networks are variants of conventional neural network models, dealing with sequential and time series data as they can be trained with some knowledge of past events. This type of model is capable of generating a sequence of outputs from a sequence of inputs and applying multiple basic operations to each input. The memory is updated step by step between the different hidden states of the network, which allows past information to be retained in order to learn temporal structures and dependencies between the data over time [58]. This type of network is widely used in natural language processing and speech recognition tasks.

Figure 4 shows the internal scheme of operation of the recurrent neural networks. For a time instant *t*, the neural network takes the input sequence $x_{t}$ and the corresponding memory vector of the immediately preceding instant $h_{t-1}$. Through a series of simple mathematical operations within the cell $C_{t}$, it obtains an output vector *y* and the updated memory state $h_{t}$. The results generated at the output of the network can be calculated as follows: (1)$$h_{t}=\sigma^{h}\left(W^{x}\cdot x_{t}+W^{h}\cdot h_{t-1}\right)$$
(2)$$y_{t}=\sigma^{y}\left(W^{y}\cdot h_{t}\right)$$

The parameters of the matrices $W^{h}$, $W^{x}$ and $W^{y}$ of Equations (Equation 1) and (Equation 2) are calculated and optimised during training. They are dense matrices whose weights are modified through the neural network training progresses. In turn, the parameters $\sigma^{h}$ and $\sigma^{y}$ correspond to non-linear activation functions.

### 4.2. LSTM-Models

LSTM layer-based neural network models are a type of recurrent network with a number of modifications to prevent the gradient vanishing that conventional RNNs suffer from [59]. This type of network is also suitable for working with time series and large volumes of data [60]. As with RNNs, LSTM networks have an internal inter-cell communication system that functions as a memory to store the temporal correlations between data.

The inner workings of the LSTM networks and the composition of their cells can be seen in Figure 5. The memory vector $h_{t}$ for a given time instant *t* and the network output vector $y_{t}$ can be calculated as follows from the memory vector of the previous instant $h_{t-1}$ and the input data vector $x_{t}$:(3)$$h_{t}=\left\{\begin{array}{ccc} tanh\left(Ux_{t}+b_{t}^{h}\right) & while & t=0\\ tanh\left(Ux_{t}+Wh_{i-1}+b_{t}^{h}\right) & while & t>0 \end{array}\right.$$
(4)$$y_{i}=tanh\left(Vh_{i}+b_{t}^{h}\right)$$

The *U*, *V* and *W* parameters of Equations (Equation 3) and (Equation 4) are calculated and optimised during training. The parameters $b_{t}^{h}$ are equivalent to the bias. The parameter *U* corresponds to the weight matrix relating the input layer to the hidden layer. The parameter *V* corresponds to the weight matrix that relates the hidden layer to the output layer. Finally, the parameter *W* corresponds to the weight matrix that relates each of the hidden layers of the LSTM network.

The main calculations that are carried out inside an LSTM cell are shown below [61]:(5)$$f_{t}=\sigma \left(U_{f}x_{t}+W_{f}h_{t-1}+b_{f}\right)$$

Equation (Equation 5) indicates how the forget gate value is calculated. Both the $U_{f}$ parameter and the $W_{f}$ parameter are calculated and modified during network training.
(6)$$i_{t}=\sigma \left(U_{i}x_{t}+W_{i}h_{t-1}+b_{i}\right)$$

Similarly, Equation (Equation 6) indicates the method for calculating the input gate value. The sub-indices of the weight matrices indicate which values corresponding to the input gate are to be used.
(7)$$g_{t}=tanh\left(U_{g}x_{t}+W_{g}h_{t-1}+b_{g}\right)$$

Equation (Equation 7) indicates how the value of $g_{t}$ is calculated. This parameter represents the value that symbolises the candidate state as being part of the LSTM memory.
(8)$$o_{t}=\sigma \left(U_{o}x_{t}+W_{o}h_{t-1}+b_{o}\right)$$

Equation (Equation 8) shows how the value of the output gate is calculated from the initial weight matrices specific to that gate.
(9)$$c_{t}=f_{t}\cdot c_{t-1}+i_{t}\cdot g_{t}$$

Equation (Equation 9) is used to calculate the state of the LSTM memory, having forgotten the values indicated by the forget gate and having included the values indicated by the input gate.
(10)$$h_{t}=o_{t}\cdot tanh\left(c_{t}\right)$$

Finally, Equation (Equation 10) is used to update the value of the hidden layers of the LSTM network, the output of which will be the input to the next cell.

### 4.3. GRU-Models

Models based on GRU neural networks are a variant of conventional RNN networks, quite similar to LSTM networks in that they solve the main problem of gradient vanishing, with the particularity that they are less computationally heavy than LSTM networks. This is because they reduce the number of internal parameters to modify and adjust during the training of the network, since one of the internal gates of the cells that make up the [62] network is eliminated.

Figure 6 shows the internal layout of a GRU network. The main difference with the previously mentioned LSTM networks is the reduction of one of its internal gates. In this way, GRU networks synthesise the former input gate and forget gate of LSTMs into one gate, the update gate [63]. GRU models are based on the following equations: (11)$$z_{t}=\sigma \left(W_{z}x_{t}+U_{z}h_{t-1}+b_{z}\right)$$

Equation (Equation 11) shows how to calculate $z_{t}$, which corresponds to the value of the update gate for a given time *t*. This gate decides what information from past events is kept in the memory, and the amount of information, in order to send it to subsequent cells.
(12)$$r_{t}=\sigma \left(W_{r}X_{t}+U_{r}h_{t-1}+b_{r}\right)$$

Equation (Equation 12) indicates the method to calculate $r_{t}$, a parameter that corresponds to the value of the reset gate. This gate decides which previous states contribute to updating the hidden states of the GRU network. For example, if the value is 0, the network forgets the previous state.
(13)$$h_{t}^{\prime }=tanh\left(W_{h}x_{t}+r_{t}\cdot U_{h}h_{t-1}\right)+b_{h}$$

Equation (Equation 13) shows how to calculate the parameter $h_{t}^{\prime }$, which represents the candidate for the new hidden state of the neural network.
(14)$$h_{t}=\left(1-z_{t}\right)\cdot h_{t-1}+z_{t}\cdot h_{t}^{\prime }$$

Finally, in Equation (Equation 14), the true value of $h_{t}$, the new hidden state, is calculated. It is important to interpret in the equation that this hidden state depends on the previous hidden state $h_{t-1}$. In addition, the matrices $W_{z}$, $W_{r}$, *W*, $U_{z}$, $U_{r}$, *U* and the bias parameters $b_{z}$, $b_{r}$ and $b_{h}$ are modified and redistributed during training.

### 4.4. Data Processing and Handling

The dataset contains a large number of records, provided that 2 months of daily life events have been recorded for a total of 2 users. For this reason, it is necessary to pre-process the data that are going to be provided to the neural network. In addition, it is necessary to note that the database records sensor events, which can lead to short periods of time with multiple events, and long intervals of time in which few events occur. Therefore, in order to use these data, a sampling system was developed to check the status of each device in the non-intrusive monitoring system as a time interval (T) elapses. This feature allows the system to operate in real time. Since there are two users, developing a unique data distribution for each user is required for the activities to be separately detected. The monitoring system is composed of 3 subsystems (see Figure 3): the non-intrusive sensors, the Bluetooth positioning beacons and the activity wristbands. Looking at these systems, it is easy to interpret that the information extracted from the Bluetooth positioning beacons and activity wristbands is unique and inherent to each user. However, the data extracted from non-intrusive sensors is common to all residents of the house, as data are extracted about events as they occur and are not linked to any one user. Therefore, each user shares the information from the sensor installation and uses their own information from the beacons and wristbands.

With respect to non-intrusive sensor sampling, it is necessary to analyse the information published by these sensors throughout the day in order to extract the main features. For example, motion sensors provide a “true” or “false” value depending on whether presence is detected or not. The same happens with contact sensors, as they provide an “open” or “closed” value. Those values need to be transformed into numerical values. Therefore, the sampling system is responsible for assigning a value of 0 or 1, according to the string of information coming from each sensor. With respect to sensors that provide a numerical value, such as temperature, consumption or lighting sensors, a normalisation from 0 to 1 has been carried out with respect to the maximum and minimum value within the measurement range of each sensor. Therefore, for motion sensors (M), the main feature is the detection of motion in a room. For door sensors (C), it is important to know whether a door is opened or closed. For vibration sensors (V), it is important to know if a furniture has received an impact or has been touched. Finally, for temperature and humidity sensors (TH), power consumption sensors (P) and light intensity sensors (L), it is important to know the temporal variations of their measurements.

Another important aspect when using the information provided by the sensors is to know the nature of the signal to be analysed. For example, to use the signal from the washing machine consumption sensor, it is only necessary to know the time at which the appliance begins to consume, and it is not so necessary to know the number of watts it is consuming or how long it takes to detect the activity of washing clothes. For this reason, a zero order retainer has been included to keep the signal at a value of 1 for a certain time range and then this value returns to 0.

Moreover, there are activities that are performed more frequently at certain times of the day. For example, the usual time to sleep is at night. Therefore, the time of day has been included as input data to the neural network, separated in normalised sine-cosine format from 0 to 1 in order to respect the numerical distances due to the cyclical nature of the measurement. This time reinforces the prediction of activities that are always performed at the same times and, in addition, helps the neural network to learn the user’s habits. Equations (Equation 15) and (Equation 16) show the calculations of the $Hour_{X}$ and $Hour_{Y}$ components as a function of the hour of the day *h* and the minute min.
(15)$$Hour_{X}=cos\left(\frac{2\pi (h+\frac{min}{60})}{24}\right)$$
(16)$$Hour_{Y}=sin\left(\frac{2\pi (h+\frac{min}{60})}{24}\right)$$

To train the neural network, it is very important to have a sufficiently distributed dataset to avoid overfitting the network to the training data. Similarly, there are activities in the dataset that are performed more frequently than others. Therefore, it is necessary to distribute the activities evenly across the training, test and validation sets, randomly enough for the system to generalise correctly and apply some data augmentation techniques in order to balance the dataset. The following data augmentation techniques were applied:Oversampling: The number of occurrences per activity throughout the dataset is variable. For this reason, if the network is trained with too many samples of one type of activity, this may mean that the system tends to generate this type of output. Therefore, in order to train the network with the same number of occurrences for each type of activity, an algorithm is applied that duplicates examples from the minority class until the number of windows is equalised.Data sharing: Despite the duplication mechanism, some activities are not able to achieve high hit rates. In order to increase the range of situations in which an activity takes place, a mechanism that shares activity records between the two users has been developed. To improve the performance of neural networks, it is beneficial to train with a greater variability of cases and situations. Therefore, for activities with worse hit rates and less variability, the algorithm adds to the training subset situations experienced by the other user to improve the generalisation of the model. For example, the activities “Chores”, “Cook”, “Pet Care” and “Read” for user 2 produced low results. For this reason, the data sharing algorithm was charged to add to the user 2’s training set, different time windows corresponding to user 1’s dataset while he/she was performing these activities. Before applying the algorithm, the average success rate of these activities was 21%. Once the algorithm was applied, the success rate exceeded 80%, which represents a significant increase in the network performance.

Figure 7 shows a diagram illustrating an example of the randomised distribution system developed for each activity. Furthermore, it can be seen that the test data set is separated from training and validation in order to prove the neural network under completely realistic scenarios. Once the test has been separated from the rest of the dataset, all contiguous blocks of activity are extracted. Obviously, each activity has a different duration in time, so this is a factor to be taken into account. Once the activity blocks have been separated, a random distribution of the total number of blocks is made, grouping them in a stack. Within each activity block, events are randomised to improve the generalisation of the model. Finally, a percentage distribution of each activity window into training and validation sets is performed. In this way, all activities will have a percentage of events belonging to the same activity block distributed among each of the training subsets.

### 4.5. Resulting Neural Network Model

To perform activity recognition, a neural network model was developed and trained individually for each user, in order to obtain the weights calculated in a personalised way for each one.

Figure 8 shows the architecture of the proposed neural network. All the input data are divided in order to feed the time series analysis layer with the sensor data only, leaving the two-time components to be used only in the final dense layer.

Three different Dropout layers with different randomisation values are used at different points of the architecture to avoid model overfitting. A Batch Normalisation layer has also been used to generate packets of the same length so as to avoid the phenomenon of internal covariation. Finally, the data analysed by the time series analysis layer (which can be either a conventional RNN, an LSTM or a GRU) are added to the two components of the time of activity and passed through a dense layer with multiple neurons, which will readjust their internal weights during training to increase the accuracy of the model.

## 5. Experiments and Discussion

In order to validate the proposed system, a supervised training of the neural network was carried out using the data provided by the non-intrusive sensors and the position information of each user provided by the beacons. To this end, the total dataset was divided as follows: 70% of the database has been used for the training dataset, 10% for the validation and 20% for the test.

First, a summary of the main hyperparameters used during training is discussed. The database was sampled in event format every 2 s. Then, a sliding window was made to collect a total of 60 samples; which means that each time window has a duration of 2 min in total. The following activities were filtered out: “Other”, “Sleep”, “Out Home” and “Watch TV” because they exceeded the rest of the activities in terms of number of samples. In this way, the training is performed in a better balanced way. Concerning the number of cells in the RNN, LSTM and GRU layers, a total of 64 cells was used. In addition, the L2 regularization was introduced with a very small value (1 × 10^−6^) to improve the performance of the temporal analysis layers and the convergence. A total of 8000 internal neurons were selected for the dense layer of the model. The value chosen for the three dropout layers of the model was 0.6. Regarding the number of epochs, the early stopping technique was used to automatically stop the training when the validation loss stopped decreasing. In this way, the overfitting of the training data can be avoided, allowing an improvement in the generalisation of the system and a decrease in meaningless epoch executions. A batch size of 128 was used. The model was trained in an Intel (R) Core (TM) i9-10900K CPU@3.70 GHz/128 Gb with two RTX3090 GPUs.

Figure 9 shows the accuracy and loss curves generated during the training of the neural network for user 1. The training was performed using the RNN (a), LSTM (b) and GRU (c) layers. The training of the RNN, LSTM and GRU models was carried out for 40, 32 and 41 epochs, respectively.

The precision and loss curves generated during the training of the neural network for user 2 can be seen in Figure 10. In the same way as for user 1, the training was performed using RNN (a), LSTM (b) and GRU layers (c), maintaining the same training conditions for both users. The training of the RNN network lasted 132 epochs, while the training of the LSTM network lasted 40 epochs, and the GRU network finished in 70 epochs.

From the learning curves for both users, it can be extracted that overfitting is small, since the early stopping technique avoids this problem. It can also be seen that training is faster in the LSTM and GRU networks, given that they are more advanced and elaborated networks than conventional RNNs.

In order to evaluate the accuracy of the model, a positive result was considered if the activity appeared in a time interval before or after the prediction equal to 10 min. In this way, the transition period between activities, which is considered a fuzzy range, can be avoided.

Table 5 shows a summary of the results of the test set for each user and their neural network type. In addition, a set of metrics corresponding to each type of neural network is included. It can be deduced that for user 1, the GRU neural network provides the best results, reaching a value of 90.91% correct. However, for user 2, the neural network that achieves the highest success is the LSTM, obtaining a result of 88.29%. Additionally evident from this table is the fact that the average number of training epochs was longer for user 2 than for user 1 because the data takes longer to converge. The difference in the hit percentage may lie in the differences during the labelling phase of the activities through the database elaboration phase. Therefore, in the next steps in the experimentation, the winning models, i.e., the GRU model for user 1 and the LSTM model for user 2, are used.

Table 6 shows a summary of the statistics obtained by activity for each user. The values of precision, recall and f1-score can be seen. The recall parameter is used as an accuracy metric to evaluate each activity. From this table, it can be concluded that, for user 1, the best detected activities are “Dress”, “Laundry”, “Pet”, “Read”, “Watch TV” and “Work”, with 100% recall; while the worst detected real activity is “Dishwashing”, with 55% recall, as more activities are carried out in that room, such as “Cook” or “Make Simple Food”. For user 2, the activities with the highest hit rates are “Chores”, “Cook”, “Dishwashing”, “Dress”, “Laundry”, “Pet”, “Shower” and “Work”, with 100% recall; while the worst detected activity is “Make Simple Food”, with 52% recall.

The results obtained from the use of the test data set for user 1 and the neural network can be analysed upon the confusion matrix reported in Table 7. In this matrix, the real activity sets of the database are placed as rows, while the predictions of the neural network are placed as columns. Therefore, the values placed on the diagonal of the matrix are equivalent to accurate predictions. It can be seen that the most influential weights are placed on the diagonal of the matrix, which results in the neural network predictions being accurate. However, some important weights can be seen far from this diagonal, such as the case of the confusion between the activity “Chores” and “Out Home”. This can be attributed to the fact that user 1 usually removes the smart band during household cleaning in order to avoid damaging the bracelet with cleaning products. There is also a confusion between “Dishwashing” and “Out Home”. This can be caused by the fact that the “Out Home” activity is one of the heaviest activities in the database in terms of number of windows and variety of situations, which means that the system may be inclined to interpret this situation.

The confusion matrix corresponding to user 2 is reported in Table 8. Most of the higher percentages are located on the diagonal of the matrix, as happened for user 1, which reveals that the accuracy of the neural network is high. Moreover, the accuracy of the neural network for user 2 is lower than the accuracy of the neural network for user 1, as was discussed before, and can be seen in the matrix. For example, the neural network for user 2 confuses the activity “Read” and “Watch TV”. This has been attributed to the fact that, normally, user 2 reads in the living room, while user 1 is watching TV in the same room. There is also a confusion between “Make Simple Food” and “Cook”. This can be caused by the fact that the nature of both activities is very similar, both take place in the kitchen and both use the drawers and the fridge. The only difference is that in the “Cook” activity there is an increase in temperature and humidity in the room due to the use of the ceramic hob.

From the results obtained, the following deductions can be made:Models based on recurrent neural networks are useful for analysing discrete time series data, such as data provided by binary sensors and positions provided by beacons.In particular, the LSTM and GRU neural networks offer better results than traditional RNNs, due to the fact that they avoid the gradient vanishing problem, with a higher computational load and a greater number of internal parameters.The early stopping technique has made it possible to stop training at the right time to avoid overfitting, which would reduce the success rate at the prediction phase.Prediction inaccuracies happen mostly on time-congruent activities or activities of a similar nature.It is possible to detect individual activities in multi-user environments. If sensor data were analysed in an isolated mode, the results would fall because these data are common to both users. The technology that makes it possible to differentiate between the two users is the positioning beacons, which indicate the position of each user in the house. In most cases, this makes it possible to discriminate which user is performing the activity.

Activity recognition using this database is a challenge, as it is difficult to discern what activity is being performed by each user in real time in a house where there is also a pet, as this can cause noise in the sensor measurements. This is a difference with respect to recognition systems that use data from homes where only one person lives, as the information provided by the sensors is inherent to that user [32].

The link to the database can be found at the end of the paper, under Data Availability Statement.

## 6. Conclusions

In this paper, a database covering the daily habits of two people living in the same house, obtained by combining different types of technology, is presented. An effective method of real-time prediction of the activities performed by these users is also included. Eighteen different activities of daily living have been considered, providing relevant information about the daily habits of the users, such as taking medicines or carrying out a correct and healthy personal hygiene. This allows a certain control over the life habits of elderly people living alone without intruding into their privacy, so that unwanted or even risky situations can be prevented.

The database is based on information provided by three subsystems: an installation of non-intrusive sensors that capture events occurring within the home, a system for positioning each user by triangulation using wireless beacons, and a system for capturing physiological signals, such as heart rate or hand position, using commercially available activity wristbands. The technology used is completely wireless and easily installable, which simplifies its implementation in a larger number of homes. The database covers 2 months of two users living with a pet.

The prediction system is able to detect the activities tagged in the proposed database with an accuracy of 90.91% for user 1 and 88.29% for user 2, using information from sensors and positioning beacons. This precision is considered satisfactory, given that both users are in the house simultaneously and performing different activities at all times. The system is capable of operating in real time, allowing it to detect activities within seconds of starting, which allows it to anticipate unwanted situations in a short period of time. This prediction system is based on recurrent neural networks, comparing the performance of three types of networks: traditional RNN, LSTM and GRU. It is important to notice that each user has his or her own personalised neural network, which makes it possible to learn from the habits and way of life of each user. Finally, it is necessary to add that the system works with the time of the day, allowing the network to elaborate a personalised habit plan depending on the time of the day.

Comparing the database proposed in the present paper with similar related works, it is worth remarking that the database is elaborated for multiple users, has a wide range of activities of daily life and uses three different types of technologies. In addition, it has a large number of events and situations and covers 2 months of data.

The system proposed in this paper is useful for taking care of the daily habits of elderly people, as it is possible to monitor their behavioural patterns remotely without disturbing their privacy. Thus, with this system, a therapist is able to monitor a larger number of users than by using traditional techniques. In addition, with this system, it is possible to anticipate dangerous situations, such as if the user has not eaten or taken their medication for several days. It is also possible to monitor vital signs remotely. For example, the heart rate can be monitored, and irregularities can be seen.

A future line of work is to use the real-time recognition system to detect dangerous situations and develop an automatic alarm system for unwanted cases, based on the activities detected in this paper. Two months of data have been collected in the work carried out in this paper, but the duration will be increased in order to obtain a larger number of samples for each activity. Another future line of work is to continue incorporating data and activities to the database by analysing a larger number of households in order to capture different behavioural patterns. Future work will also focus on making the system replicable in a larger number of households. In parallel, research will continue on new artificial intelligence techniques to analyse time series data. Finally, work will be carried out on incorporating the data obtained from the activity wristbands into the prediction system to try to increase the percentage of accuracy of the neural networks.

## Figures and Tables

**Figure 1 sensors-22-08109-f001:**
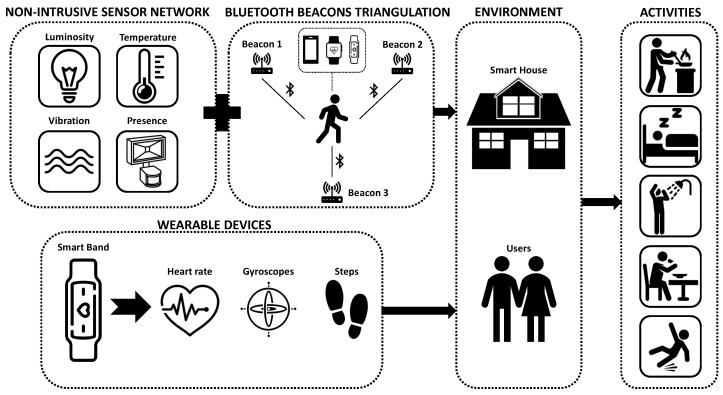
General diagram of the components of the non-intrusive monitoring system.

**Figure 2 sensors-22-08109-f002:**
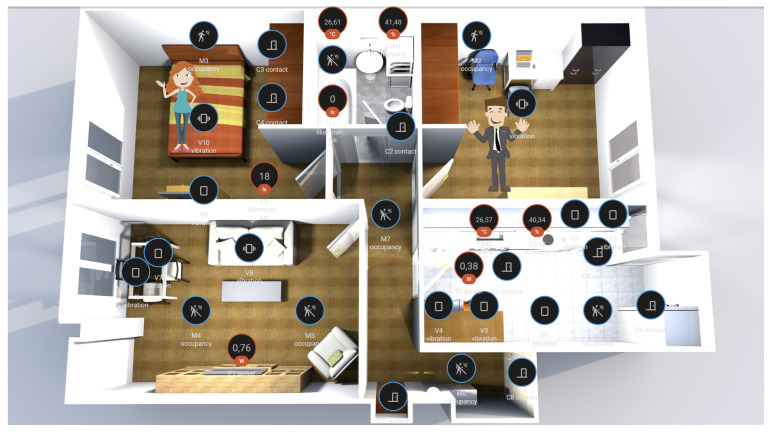
Visual interface provided by the Home Assistant operating system.

**Figure 3 sensors-22-08109-f003:**
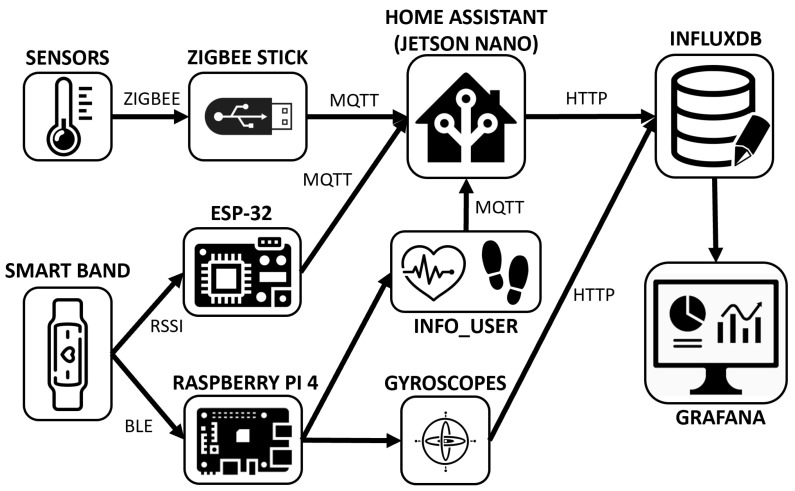
Communication diagram of the components of the non-intrusive monitoring system.

**Figure 4 sensors-22-08109-f004:**
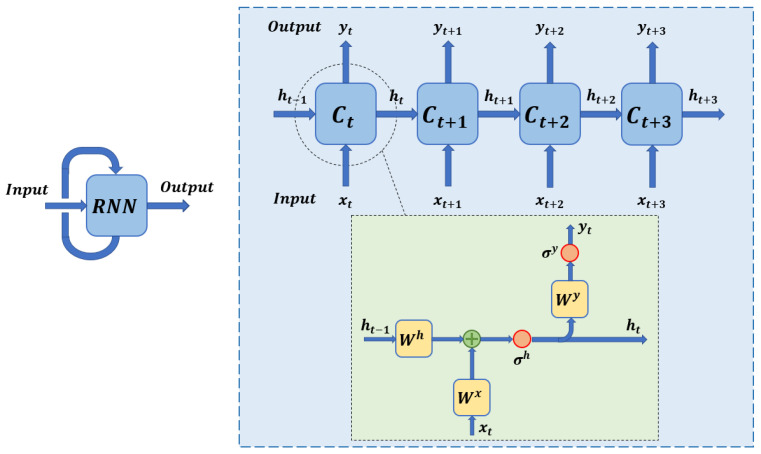
Generic schema of the content of a recurrent neural network model.

**Figure 5 sensors-22-08109-f005:**
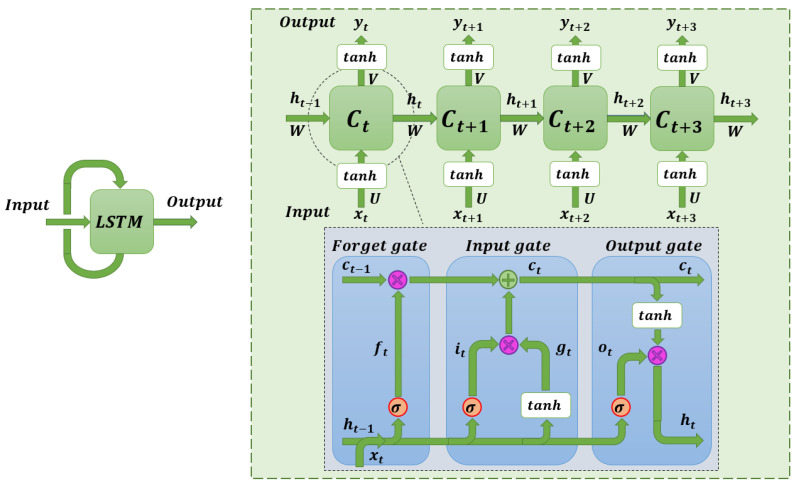
Generic schema of the content of an LSTM neural network model.

**Figure 6 sensors-22-08109-f006:**
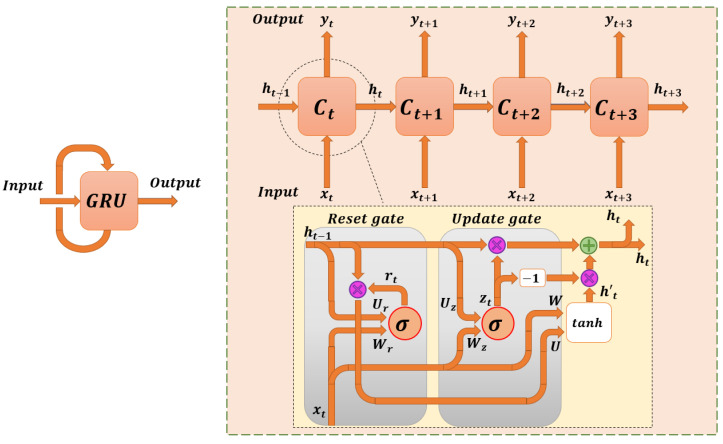
Generic schema of the content of a GRU neural network model.

**Figure 7 sensors-22-08109-f007:**
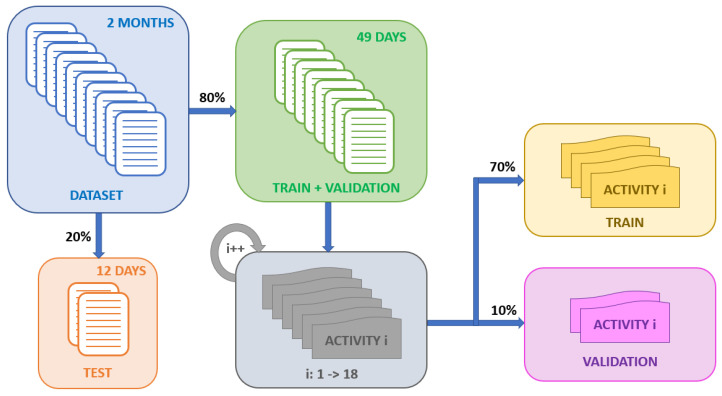
Example of the randomised distribution process.

**Figure 8 sensors-22-08109-f008:**
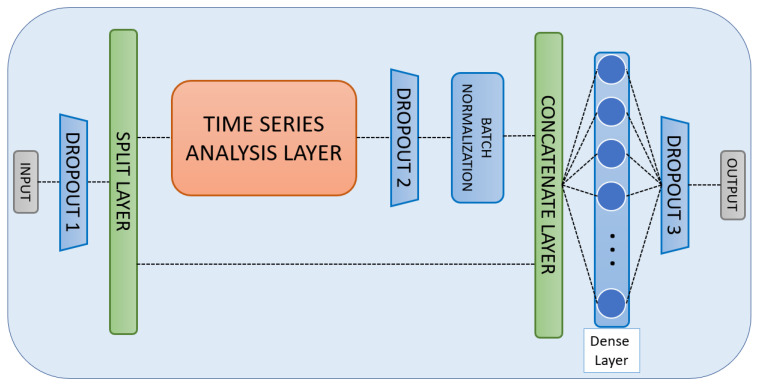
Neural network architecture model for both users.

**Figure 9 sensors-22-08109-f009:**
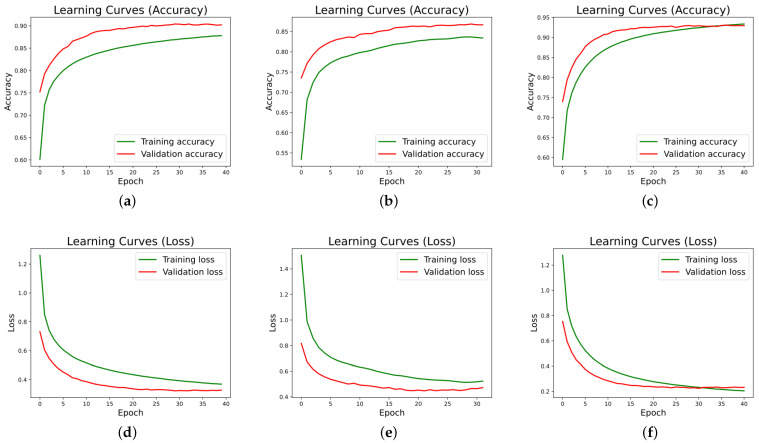
Neural network training graphs for user 1. (**a**) Model RNN Accuracy User 1, (**b**) Model LSTM Accuracy User 1, (**c**) Model GRU Accuracy User 1, (**d**) Model RNN Loss User 1, (**e**) Model LSTM Loss User 1, (**f**) Model GRU Loss User 1.

**Figure 10 sensors-22-08109-f010:**
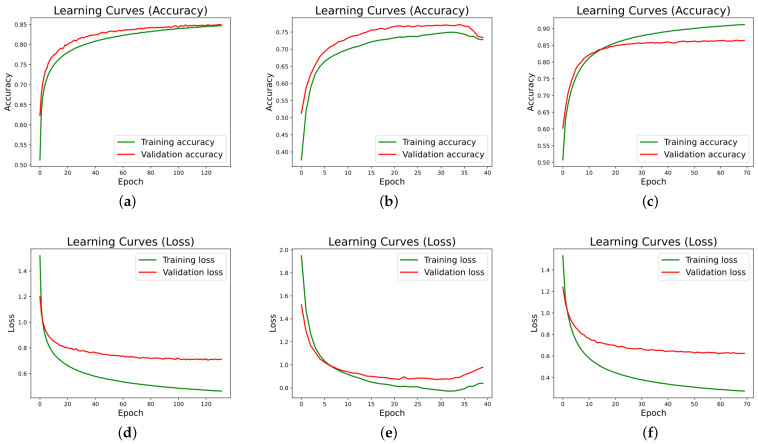
Neural network training graphs for user 2. (**a**) Model RNN Accuracy User 2, (**b**) Model LSTM Accuracy User 2, (**c**) Model GRU Accuracy User 2, (**d**) Model RNN Loss User 2, (**e**) Model LSTM Loss User 2, (**f**) Model GRU Loss User 2.

**Table 1 sensors-22-08109-t001:** Overview of major databases on activities of daily living.

Name of Database	Houses	Multi-Person	Duration	Type and Number of Sensors	Activities
USC-HAD [16]	1	No	6 h	Wearable sensors (5)	12
MIT PlaceLab [17]	2	No	2–8 months	Wearable + Ambient sensors (77–84)	10
ContextAct@A4H [18]	1	No	1 month	Ambient sensors + Actuators (219)	7
SIMADL [19]	1	No	63 days	Ambient sensors (29)	5
MARBLE [20]	1	No	16 h	Wearable + Ambient Sensors (8)	13
OPPORTUNITY [21]	15	No	25 h	Wearable + Ambient sensors (72)	18
UvA [22]	3	No	28 days	Ambient sensors (14)	10–16
CSL-SHARE [23]	1	No	2 h	Wearable sensors (10)	22
NTU RGB+D [24]	-	Yes	-	RGB-D cameras	60
ARAS [25]	2	Yes	2 months	Ambient sensors (20)	27
CASAS [26]	7	Yes	2–8 months	Wearable + Ambient sensors (20–86)	11
ADL [29]	1	No	2 h	Wearable sensors (6)	9
Cogent-House [30]	1	No	23 min	Wearable sensors (12)	11
Pires, I. et al. [31]	1	No	14 h	Smartphone sensors	5
SDHAR-HOME (Proposed)	1	Yes	2 months	Wearable + Ambient sensors (35) + Positioning (7)	18

**Table 2 sensors-22-08109-t002:** Set of activities labelled in the database.

	Activities	
Bathroom Activity	Chores	Cook
Dishwashing	Dress	Eat
Laundry	Make Simple Food	Out Home
Pet Care	Read	Relax
Shower	Sleep	Take Meds
Watch TV	Work	Other

**Table 3 sensors-22-08109-t003:** Study of sensors needed to detect the proposed activities.

Room	Activity	Sensors	Location
Bedroom	Sleep	PIR	Wall
		Vibration	Bed
		Light	Wall
	Dress	PIR	Wall
		Contact	Wardrobe
		Vibration	Wardrobe
	Read	PIR	Wall
		Light	Wall
Bathroom	Take Meds	PIR	Wall
		Contact	Drawer
	Bathroom Activity	PIR	Wall
		Light	Wall
	Shower	PIR	Wall
		Light	Wall
		Temp.+Hum.	Wall
Kitchen	Cook	PIR	Wall
		Temp.+Hum.	Wall
		Contact	Appliances
	Dishwashing	PIR	Wall
		Contact	Dishwasher
	Make Simple Food	PIR	Wall
		Contact	Wardrobes
	Eat	PIR	Wall
		Light	Wall
		Vibration	Chair
	Pet Care	PIR	Wall
		Vibration	Bowl
	Laundry	PIR	Wall
		Consumption	Washer
Study Room	Work	PIR	Wall
		Vibration	Chair
Lounge	Watch TV	PIR	Wall
		Consumption	TV
		Vibration	Sofa
	Relax	PIR	Wall
		Vibration	Sofa
Hall	Out Home	PIR	Wall
		Contact	Door
	Chores	PIR	Wall
		Contact	Wardrobe

**Table 4 sensors-22-08109-t004:** Overview of sensors in the installation: types and references.

Type of Sensor	Reference	Total Number
Aqara Motion Sensor (M)	RTCGQ11LM	8
Aqara Door and Window Sensor (C)	MCCGQ11LM	8
Aqara Temperature and Humidity Sensor (TH)	WSDCGQ11LM	2
Aqara Vibration Sensor (V)	DJT11LM	11
Xiaomi Mi ZigBee Smart Plug (P)	ZNCZ04LM	2
Xiaomi MiJia Light Intensity Sensor (L)	GZCGQ01LM	2
		TOTAL = 33

**Table 5 sensors-22-08109-t005:** Summary of neural network training for both users.

User	Metrics	RNN Network	LSTM Network	GRU Network
User 1	Accuracy	89.59%	89.63%	90.91%
	Epochs	40	32	41
	Training time	18,222 s	35,616 s	56,887 s
	Test time	32 s	47 s	110 s
	Parameters	1,515,410	2,508,050	2,177,554
User 2	Accuracy	86.26%	88.29%	86.21%
	Epochs	132	40	70
	Training time	67,115 s	50,209 s	111,223 s
	Test time	32 s	94 s	110 s
	Parameters	1,515,410	2,508,050	2,177,554

**Table 6 sensors-22-08109-t006:** Summary table of results by activity (User 1–User 2).

Activity	Precision	Recall	F1-Score
Bathroom Activity	0.87–0.82	0.92–0.99	0.89–0.90
Chores	0.99–0.64	0.59–1.00	0.74–0.78
Cook	0.96–0.72	0.79–1.00	0.86–0.84
Dishwashing	0.64–1.00	0.55–1.00	0.59–1.00
Dress	0.94–0.40	1.00–1.00	0.97–0.57
Eat	0.97–0.97	0.87–0.95	0.92–0.96
Laundry	1.00–0.49	1.00–1.00	1.00–0.66
Make Simple Food	0.98–0.80	0.92–0.52	0.95–0.63
Out Home	0.94–1.00	0.96–0.91	0.95–0.95
Pet	0.97–0.88	1.00–1.00	0.98–0.93
Read	0.91–0.58	1.00–0.54	0.95–0.56
Relax	0.42–0.82	0.91–0.84	0.58–0.83
Shower	0.85–0.98	0.86–1.00	0.85–0.99
Sleep	1.00–0.92	0.83–0.87	0.90–0.89
Take Meds	1.00–0.76	0.93–0.94	0.97–0.84
Watch TV	0.94–0.69	1.00–0.94	0.97–0.80
Work	0.99–0.95	1.00–1.00	1.00–0.97
Other	0.53–0.70	0.93–0.75	0.68–0.72
Accuracy			0.91–0.88
Macro avg.	0.88–0.78	0.89–0.90	0.88–0.82
Weighted avg.	0.93–0.92	0.91–0.88	0.91–0.90

**Table 7 sensors-22-08109-t007:** Neural network User 1: Confusion Matrix.

		Predicted																	
		**Bathroom Activity**	**Chores**	**Cook**	**Dishwashing**	**Dress**	**Eat**	**Laundry**	**Make Simple Food**	**Out Home**	**Pet**	**Read**	**Relax**	**Shower**	**Sleep**	**Take Meds**	**Watch TV**	**Work**	**Other**
**Actual**	**Bathroom Activity**	0.92	0	0	0	0	0	0	0	0.05	0	0	0	0.02	0	0	0	0	0.01
	**Chores**	0	0.59	0	0	0	0	0	0	0.35	0	0	0.04	0	0	0	0	0	0.01
	**Cook**	0	0	0.79	0.07	0	0.10	0	0	0	0	0	0	0	0	0	0.05	0	0
	**Dishwashing**	0	0	0	0.55	0	0.12	0	0	0.26	0	0	0	0	0	0	0.07	0	0
	**Dress**	0	0	0	0	1.00	0	0	0	0	0	0	0	0	0	0	0	0	0
	**Eat**	0.03	0	0	0	0	0.87	0	0	0.01	0	0.01	0.02	0	0	0	0.06	0	0
	**Laundry**	0	0	0	0	0	0	1.00	0	0	0	0	0	0	0	0	0	0	0
	**Make Simple Food**	0	0	0	0	0	0	0	0.92	0.03	0	0	0	0	0	0	0.05	0	0
	**Out Home**	0	0	0	0	0	0	0	0	0.96	0	0	0.04	0	0	0	0	0	0
	**Pet**	0	0	0	0	0	0	0	0	0	1.00	0	0	0	0	0	0	0	0
	**Read**	0	0	0	0	0	0	0	0	0	0	1.00	0	0	0	0	0	0	0
	**Relax**	0	0	0	0	0	0	0	0	0	0	0	0.91	0	0	0	0.09	0	0
	**Shower**	0.14	0	0	0	0	0	0	0	0	0	0	0	0.86	0	0	0	0	0
	**Sleep**	0.01	0	0	0	0	0	0	0	0.05	0	0	0	0	0.83	0	0	0	0.11
	**Take Meds**	0.01	0	0	0	0	0	0	0	0	0	0	0.01	0	0	0.93	0	0	0.05
	**Watch TV**	0	0	0	0	0	0	0	0	0	0	0	0	0	0	0	1.00	0	0
	**Work**	0	0	0	0	0	0	0	0	0	0	0	0	0	0	0	0	1.00	0
	**Other**	0	0	0.01	0	0	0.01	0	0	0	0	0.02	0.01	0	0.01	0	0.01	0	0.93

**Table 8 sensors-22-08109-t008:** Neural network User 2: Confusion Matrix.

		Predicted																	
		**Bathroom Activity**	**Chores**	**Cook**	**Dishwashing**	**Dress**	**Eat**	**Laundry**	**Make Simple Food**	**Out Home**	**Pet**	**Read**	**Relax**	**Shower**	**Sleep**	**Take Meds**	**Watch TV**	**Work**	**Other**
**Actual**	**Bathroom Activity**	0.99	0	0	0	0.01	0	0	0	0	0	0	0	0	0	0	0	0	0
	**Chores**	0	1.00	0	0	0	0	0	0	0	0	0	0	0	0	0	0	0	0
	**Cook**	0	0	1.00	0	0	0	0	0	0	0	0	0	0	0	0	0	0	0
	**Dishwashing**	0	0	0	1.00	0	0	0	0	0	0	0	0	0	0	0	0	0	0
	**Dress**	0	0	0	0	1.00	0	0	0	0	0	0	0	0	0	0	0	0	0
	**Eat**	0	0	0	0	0	0.95	0	0	0	0	0	0	0	0	0	0.05	0	0
	**Laundry**	0	0	0	0	0	0	1.00	0	0	0	0	0	0	0	0	0	0	0
	**Make Simple Food**	0	0	0.41	0	0	0.01	0	0.52	0	0	0.01	0	0	0.01	0	0.01	0.03	0
	**Out Home**	0	0	0	0	0.04	0	0	0	0.91	0	0	0.01	0	0.04	0	0	0	0
	**Pet**	0	0	0	0	0	0	0	0	0	1.00	0	0	0	0	0	0	0	0
	**Read**	0.01	0.02	0	0	0	0	0	0	0	0	0.54	0	0	0	0	0.42	0	0
	**Relax**	0	0.01	0	0	0	0	0	0	0	0	0	0.84	0	0	0	0.15	0	0
	**Shower**	0	0	0	0	0	0	0	0	0	0	0	0	1.00	0	0	0	0	0
	**Sleep**	0.01	0	0	0	0.03	0	0	0	0	0	0.02	0	0	0.87	0	0.02	0	0.05
	**Take Meds**	0.02	0	0	0	0.04	0	0	0	0	0	0	0	0	0	0.94	0	0	0
	**Watch TV**	0	0	0	0	0	0	0.01	0	0	0	0	0.04	0	0	0	0.94	0	0
	**Work**	0	0	0	0	0	0	0	0	0	0	0	0	0	0	0	0	1.00	0
	**Other**	0.01	0.01	0.01	0	0	0.01	0.01	0.01	0	0	0.01	0.06	0	0.13	0	0	0	0.75

## Data Availability

Publicly available datasets have been developed in this study. These data can be found here: https://github.com/raugom13/SDHAR-HOME-A-Sensor-Dataset-for-Human-Activity-Recognition-at-Home.

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
