# Peer review of "SDHAR-HOME: A Sensor Dataset for Human Activity Recognition at Home"

_sensors, 2022, doi:10.3390/s22218109_

Round 1

Reviewer 1 Report

This is a research paper on the activity recognition of multi-user using multi-sensor.

Recently, research on recognizing human activity has been actively conducted

Many researchers in the industry are interested in this field.

The authors claim that the proposed method is new and superior to the conventional methods.

However, based on the following reasons, I think it is insufficient to publish, and I recommend the "reject" decision because a lot of modifications are needed overall.

1. Lack of comparative analysis with previous studies

- Although related works (ARAS, CASAS) mentioned in the paper have stated that they are less accurate than the proposed method, they are merely claims and fail to present experimental results or objective evidence.

- Difficult to find differentiation compared to the following latest studies in terms of the problems and solutions to be solved.

1) Human activity recognition with smartphone sensors using deep learning neural networks

https://doi.org/10.1016/j.eswa.2016.04.032

2) Deep Learning for Sensor-based Human Activity Recognition: Overview, Challenges, and Opportunities

https://doi.org/10.1145/3447744

3) Deep Learning Models for Real-time Human Activity Recognition with Smartphones

https://link.springer.com/article/10.1007/s11036-019-01445-x

4) Human Action Recognition From Various Data Modalities: A Review

https://ieeexplore.ieee.org/abstract/document/9795869

5) Deep Learning in Human Activity Recognition with Wearable Sensors: A Review on Advances

https://www.mdpi.com/1424-8220/22/4/1476

6) Human Activity Recognition via Hybrid Deep Learning Based Model

https://www.mdpi.com/1424-8220/22/1/323

7) Human Activity Recognition Based on Residual Network and BiLSTM

https://www.mdpi.com/1424-8220/22/2/635

8) Human Activity Recognition Using an Ensemble Learning Algorithm with Smartphone Sensor Data

https://www.mdpi.com/2079-9292/11/3/322

9) Real-Time Human Activity Recognition Using Conditionally Parametrized Convolutions on Mobile and Wearable Devices

https://ieeexplore.ieee.org/abstract/document/9705554

10) A Lightweight Model for Human Activity Recognition Based on Two-Level Classifier and Compact CNN Model

https://link.springer.com/chapter/10.1007/978-3-030-70601-2_276

2. Unrealistic assumptions and modeling

- It is meaningless to distinguish users wearing wearable devices (e.g. Smartband, smartwatch, etc.) by sensor data because it is clearly distinguished by the ID and personal information of the wearable device, isn't it? As a result, the multi-user classification claimed in this paper seems to be trivial.

- Multi-sensor will have different sensing performance depending on channel interference, movement of moving objects, and network environment conditions. But, no evaluation of this part.

- No description of how it can distinguish between two residents and intermittent visitors. Can the accumulated two resident data distinguish intermittent visitors? How?

- How do you deal with sensor noise when residents and pets live together?

3. Unscientific and disorganized experiments

- LSTM, GRU, and RNN need to be compared and analyzed from a multilateral perspective because the advantages and disadvantages of the learning model are clear and the difference in complexity and latency as well as accuracy is clear.

- To demonstrate that it is possible to detect emergencies in elderly people living alone mentioned in the introduction of the paper, it must be an experimental environment that accurately detects emergencies, not the daily experimental conditions.

- It is said that activity can be estimated without privacy exposure, but there is a lack of proof. The privacy protection technology of the dataset (e.g. differential privacy, etc.) is not applied, and if the experimental environment is limited to two residents, it is one of two people and can be easily identified based on sensor data.

- The user's activity can be estimated even in a way other than the deep learning approach. These two methods should be compared in terms of complexity, latency, and accuracy.

- Multi-user means two or more people, and this paper shows the evaluation results only for two people.

The title must be changed to "A Two User Sensor...", or N users (N=3,4,5,6...) must be tested for Multi-user.

- If some of the multi-sensors used in this experiment are used, how does the performance come out? Which of the sensors you use is dominant? There are no explanations and experiments for these obvious questions.

4. Lack of specific explanation.

- Using the proposed system, the authors claim the proposed system could predict situations where a person "took" or "did not take medicine", so how can you classify "simple Eat" and "drug use"? And it is necessary to explain how to accurately recognize similar behaviors such as "cook" and "make simple food".

- There are many sensors used in the proposed non-intrusive monitoring system, and there is no specific explanation on how each sensor is applied to activity recognition, but it is explained that it simply learns data in deep learning to produce results.

- Why are the sensors used necessary, and what features were used for learning? And what activity is perceived by what principle?

- There is no specific description and evaluation of the data sharing algorithm mentioned in the abstract.

5. Minor comments

- The resolution of Figure 2 needs to be improved.

- Abstract and introduction of the paper lack relevance to the subject of the paper.

- The learning model description in the text is at the level of general textbook content and does not need to be explained at length. Instead, the authors should explain the proposed method for solving the problem defined in the paper.

Author Response

Dear reviewer,
Thank you very much for your valuable suggestions. Please find attached the comments to your suggestions below.

Reviewer 2 Report

In this paper, the authors present a dataset for human activity recognition. From this data, they develop a personalized prediction model with accuracy between 88.29% and 90.91% which could serve as a baseline.

The paper would benefit from thorough checking of grammar and style. Also, a lot of terms should be explained (Grafana, MQTT...)

Some of the references in Table 1 are some years old and you could mention other datasets even if they are not directly comparable with yours or your research. It could be interesting to acknowledge that such datasets exist and why you do not want to compare with them. For example, you could justify that some dataset is built from actions performed by volunteers/actors, not by people as they go in their daily lives. I attach some references, but I am sure others can be found.

Lago, P., Lang, F., Roncancio, C., Jiménez-Guarín, C., Mateescu, R., Bonnefond, N. (2017). The ContextAct@A4H Real-Life Dataset of Daily-Living Activities. In: Brézillon, P., Turner, R., Penco, C. (eds) Modeling and Using Context. CONTEXT 2017. Lecture Notes in Computer Science(), vol 10257. Springer, Cham. https://doi.org/10.1007/978-3-319-57837-8_14

Alshammari, Talal, Nasser Alshammari, Mohamed Sedky, and Chris Howard. 2018. "SIMADL: Simulated Activities of Daily Living Dataset" Data 3, no. 2: 11. https://doi.org/10.3390/data3020011

M. Saleh, M. Abbas and R. B. Le Jeannès, "FallAllD: An Open Dataset of Human Falls and Activities of Daily Living for Classical and Deep Learning Applications," in IEEE Sensors Journal, vol. 21, no. 2, pp. 1849-1858, 15 Jan.15, 2021, doi: 10.1109/JSEN.2020.3018335.

Ruzzon, Marco, Alessandro Carfì, Takahiro Ishikawa, Fulvio Mastrogiovanni, and Toshiyuki Murakami. "A multi-sensory dataset for the activities of daily living." Data in brief 32 (2020): 106122.

Olukunle Ojetola, Elena Gaura, and James Brusey. 2015. Data set for fall events and daily activities from inertial sensors. In Proceedings of the 6th ACM Multimedia Systems Conference (MMSys '15). Association for Computing Machinery, New York, NY, USA, 243–248. https://doi.org/10.1145/2713168.2713198

Pires, Ivan Miguel, Nuno M. Garcia, Eftim Zdravevski, and Petre Lameski. "Activities of daily living with motion: A dataset with accelerometer, magnetometer and gyroscope data from mobile devices." Data in brief 33 (2020): 106628.

Arrotta, L., Bettini, C., Civitarese, G. (2022). The MARBLE Dataset: Multi-inhabitant Activities of Daily Living Combining Wearable and Environmental Sensors Data. In: Hara, T., Yamaguchi, H. (eds) Mobile and Ubiquitous Systems: Computing, Networking and Services. MobiQuitous 2021. Lecture Notes of the Institute for Computer Sciences, Social Informatics and Telecommunications Engineering, vol 419. Springer, Cham. https://doi.org/10.1007/978-3-030-94822-1_25

H. Pirsiavash and D. Ramanan, "Detecting activities of daily living in first-person camera views," 2012 IEEE Conference on Computer Vision and Pattern Recognition, 2012, pp. 2847-2854, doi: 10.1109/CVPR.2012.6248010.

A. Cartas, P. Radeva and M. Dimiccoli, "Activities of Daily Living Monitoring via a Wearable Camera: Toward Real-World Applications," in IEEE Access, vol. 8, pp. 77344-77363, 2020, doi: 10.1109/ACCESS.2020.2990333.

Sucerquia, Angela, José David López, and Jesús Francisco Vargas-Bonilla. 2017. "SisFall: A Fall and Movement Dataset" Sensors 17, no. 1: 198. https://doi.org/10.3390/s17010198

Section 2.2 might be expanded and you could point the reader to some review on activity recognition. For example:

Rodríguez-Moreno, Itsaso, José María Martínez-Otzeta, Basilio Sierra, Igor Rodriguez, and Ekaitz Jauregi. 2019. "Video Activity Recognition: State-of-the-Art" Sensors 19, no. 14: 3160. https://doi.org/10.3390/s19143160

Yadav, Santosh Kumar, Kamlesh Tiwari, Hari Mohan Pandey, and Shaik Ali Akbar. "A review of multimodal human activity recognition with special emphasis on classification, applications, challenges and future directions." Knowledge-Based Systems 223 (2021): 106970.

Chen, Kaixuan, Dalin Zhang, Lina Yao, Bin Guo, Zhiwen Yu, and Yunhao Liu. "Deep learning for sensor-based human activity recognition: Overview, challenges, and opportunities." ACM Computing Surveys (CSUR) 54, no. 4 (2021): 1-40.

Sections 4.1, 4.2, and 4.3 could be reduced.

When the reader arrives at Figure 1 then realizes what the sensors exactly are. It would be helpful to introduce them before. For example, "beacons" are mentioned, but not the origin of signals that they capture and process.

Line 30: what do you mean by "the amount of activities"? I understood "the number of different activities", but in the next sentence, you only talk about the time devoted to each activity. I do not see how the sentences in lines 32-35 ("For example... activities") fit in the middle of the preceding and following sentences.

Line 38: "this remains a challenge". Is it a challenge to publish datasets, as current datasets present limitations? In lines 39-40 there is a challenge that seems more related to the problem of activity recognition, not to dataset generation.

Line 45: "peopole's".

Line 53: "provides" -> "provide".

Line 77: "persevering" -> "preserving".

Table 1: "This is a wide table".

Line 201: Please state that what a InfluxDB database is is going to be explained later.

Lines 213-215: Has the user to tag all her daily activities? Is the user doing that faithfully? I am not sure this is a non-intrusive method if the user has to be all-time alert of doing it right.

Line 300: why "these sensors are common" implies "do not provide information about which user is performing the activity".

Line 370: "decribed" -> "described".

One of my main concerns is how accurate is the tagging of the daily activities if the user has to do it all the time. Do you have any way of knowing if the tagging is right? And, are the two people in the house elderly? Because you mention the interest of this research for monitoring elderly people, but I am not sure if the data of this database comes from elderly people.

Author Response

(The authors gave the same response as above.)

Reviewer 3 Report

In my opinion, the article is well presented. The work is meticulously done, and state of the art is enriched with a new dataset.

My only suggestion is to enrich the state of the art when citing CASAS, which is a very important dataset in the field. Some examples are as follows:

1) https://www.sciencedirect.com/science/article/pii/S0925231219304862

2) https://ieeexplore.ieee.org/abstract/document/9861155/

3) https://link.springer.com/article/10.1007/s12559-020-09816-3

I suggest using them to explore the main challenges of this kind of task, in order to make the work clearer to the audience.

Best regards

Author Response

(The authors gave the same response as above.)

Round 2

Reviewer 2 Report

I believe my main concerns have been addressed.